# Emotional Intelligence in Ultra-Marathon Runners: Implications for Recovery Strategy and Stress Responses during an Ultra-Endurance Race

**DOI:** 10.3390/ijerph19159290

**Published:** 2022-07-29

**Authors:** Michel Nicolas, Marvin Gaudino, Virginie Bagneux, Gregoire Millet, Sylvain Laborde, Guillaume Martinent

**Affiliations:** 1Laboratory Psy-DREPI (EA 7458), University of Bourgogne Franche-Comté, 21000 Dijon, France; marvin.gaudino@u-bourgogne.fr; 2LPCN, Université de Caen Normandie, 14032 Caen, France; virginie.bagneux@unicaen.fr; 3SSUL, Institute of Sport Sciences, Faculty of Biology and Medicine, University of Lausanne, CH-1011 Lausanne, Switzerland; gregoire.millet@unil.ch; 4Department of Performance Psychology, Institute of Psychology, German Sport University Cologne, 50923 Cologne, Germany; s.laborde@dshs-koeln.de; 5Laboratory on Vulnerabilities and Innovation in Sport, University of Lyon 1, 69367 Lyon, France; guillaume.martinent@univ-lyon1.fr

**Keywords:** emotional intelligence, recovery stress states, mountain ultra-marathon

## Abstract

The aim of this research was to investigate the role of trait emotional intelligence (EI) in recovery stress states in a mountain ultra-marathon (MUM) race. Recovery stress states of 13 finishers were assessed before, during, and immediately after the end of an extreme MUM, whereas emotional intelligence was assessed 2 days before the MUM race. Temporal evolutions of recovery stress states were examined. Stress states increased after the race whereas recovery states decreased in all participants. In addition, recovery states were influenced by the trait EI level assessed before the competition. Results supported the hypothesis that trait EI tends to have a positive effect by boosting recovery strategies. In this perspective, trait EI could have a protective role against stress and improve pre-competition mental preparation. High scores of trait EI (in comparison to low scores of trait EI) could have helped athletes to increase recovery states in order to improve their psychological adaptation to one of the most difficult races in the world.

## 1. Introduction

Extreme sports situations demand multidimensional psychological adaptive responses which could depend on recovery stress states [1], as well as individual factors such as emotional intelligence [2]. Biopsychological perspective of recovery and stress [3], embraces physical and biopsychosocial dimensions of both stress and recovery to indicate the extent to which someone is physically and/or mentally stressed, as well as whether that person is capable of using individual strategies for recovery and which strategies are used. EI refers to a form of intelligence which aims to capture individual differences in interpersonal and intrapersonal emotional functioning [4,5,6]. Its potential contribution in sporting competitions has been demonstrated and is considered to be a key factor in improving individual adaptation, notably with regard to the stress process [2]. During the last few decades, the recovery process has been associated with stress states to explain how athletes may be better able to tolerate and buffer stress from training and competition [7]. Whereas the relationship between stress and EI has been largely documented, no study has investigated the relationship between EI and recovery. The aim of this paper is to evaluate the involvement of EI in recovery stress states.

Extreme situations are demanding and challenging. They impose on an individual the need to cope with exceptional physical or psychosocial circumstances that require adaptive responses that engage personal resources which could be overwhelmed [8]. A mountain ultra-marathon (MUM) race could be considered one of the most extreme sporting situations after polar expeditions [9] because it implies a complex and multidimensional adaptation defined by the dynamic impact between environmental and personal constraints and resources (i.e., physical, psychological, and social) on adjustment [8,10]. During a MUM, athletes are exposed to a variety of stressors and have to run for extended periods over long distances and dangerous terrain with changing altitudes in an uncertain and risky environment [11]. Exposure to these stressful environmental and climatic conditions tends to push participants to draw on their own resources in order to perform beyond their ordinary limits [12]. MUM is by definition a playing field for in situ ecological research investigating psychological impairments that are mirrored in multidimensional psychological processes, such as emotional disturbances [13] and increases/decreases in recovery stress states [14]. These impairments were also observed during the month following the competition, highlighting that ultra-endurance sports are challenging situations with long-term repercussions [14,15]. 

According to Lazarus and Folkman, the seminal model of psychological stress (1984), extreme situations can exacerbate stress states [8]. Beyond a certain point, any effort to manage an excessive stress state could engage personal resources and in turn cause their potential consumption if the recovery process is not implemented [16]. However, a certain level of perceived stress is an integral part of psychological adaptation [17]. The objective is no longer to annihilate stress but to attempt to reach a balance between stress state and personal resources. The recovery process actually represents a core concept in investigating how to deal with and buffer the stress state because it helps to protect, build, refill or restore personal resources [7,18]. Recovery is defined as a multilevel process used to tolerate stress and to re-establish performance abilities and psychological and physical strength in order to optimize situational conditions [7]. Thus, recovery is based on proactive and self-initiated activities [18,19,20].

In the last decade, there has been an increased interest in the investigation of the interrelated dynamics of recovery stress states in order to better understand the psychological adaptations in extreme situations. The theoretical model of the recovery stress process [3,19] leads to a joint measurement of the extent to which an individual is frequently and multidimensionally stressed (social, emotional, physical, and behavioral) and its recovery-associated activities/states. The objective is to reach an individual biopsychosocial balance in order to counterbalance the negative effects of stress, help to adjust to the situation, and to achieve a continuous high-level performance [3]. Results from individuals’ exposure to spatial simulations [21], polar stations, i.e., wintering in Antarctica [22], and extreme sports [14] have provided strong evidence of the importance of considering the recovery stress process. Unbalanced recovery stress states (i.e., increased stress states and decreased recovery states) can lead to dysfunctional outcomes such as chronic fatigue and concomitant overtraining, and psychological exhaustion [1,20]. Consequently, the participant’s adaptation to sports training and competition is compromised [1,19]. Results of previous studies on ultra-endurance races showed that participants have simultaneously reported an increase in stress states and a decrease in recovery states mirrored, notably, in the emotional exhaustion dimension [14,15]. The repercussions could be observed up to four weeks after the race, highlighting the long-term impact of a stressful event on the recovery stress states [15]. In particular, evolutions in the recovery stress states experienced by MUM runners in the month following a demanding MUM race have been characterized by a significant linear increase in recovery and a linear decrease in stress states [15]. Results show that the harder the situation is, the longer the need to evaluate and manage recovery stress states.

However, even in extreme environments, recovery stress states are not always unbalanced [22,23], suggesting that personal resources could be sustained. Results from previous research showed that recovery stress states could be modulated according to individual difference factors such as perceived stress and perceived control [19,22,24]. Specifically, these studies have shown that perceived stress was positively linked to psychological, physiological, and social stress responses whereas perceived control was positively linked to recovery strategies [19,22,24]. These results provided supporting evidence that individual cognitive resources were involved in managing the stress process [10,17]. These results corroborate the cognitive–motivational–relational theory [10] and emphasize that the interaction between the person and the environment is mediated by the degree to which a situation is appraised as stressful and controllable. These findings suggest that individual differences could help explain the differences in psychological responses within a challenging situation [25].

Among individual variables identified as factors influencing stress management, EI could determine an athlete’s ability to handle psychological stress and also facilitate physical and psychological recovery [26]. The theoretical nature of EI-related constructs remains assigned to a wide array of concepts and models [27]. EI provides an interesting framework for assessing individual differences with regard to how individuals identify, express, understand, regulate and use their own and others’ emotions to ultimately guide their thinking and actions [4,5]. Among the several theoretical frameworks focusing on EI [5,28,29,30], the present study was grounded within the trait model of EI [29,30] based on the rationale that EI was conceptualized in the present study as an individual difference variable. The trait model [29] defines EI as a lower-order personality trait that is mainly evaluated using a self-report measure [30].

A systematic review [2] concluded that EI had a protective role with regard to the stress process in athletes. For example, EI was found to be associated with the use of more efficient coping strategies (i.e., task-oriented coping) to manage stress [31]. Furthermore, Laborde, Dosseville, Guillén, and Chavez [32] indicated that EI positively predicted perceived control, coping (e.g., task-oriented coping strategies, coping effectiveness), and performance satisfaction. In addition, numerous studies support the idea that EI is a key factor in improving individual adaptation [33,34]. Individuals with high EI would be more competent in coping with challenges and would perceive less stress [35] and more well-being [36]. EI may help to explain how stress is physiologically better tolerated and buffered by certain individuals [37]. Previous research within the context of MUM race also supported the notion that EI is positively associated with pleasant emotional states [38]. The connection between EI and pleasant emotions could be crucial to depicting the relationship between EI and the recovery process. In her broaden-and-build theory, Fredrickson [39] posited that experiences of pleasant emotions broaden people’s momentary thought–action repertoires in a way that serves to build their enduring personal resources and subsequent emotional well-being. Several empirical studies provide evidence supporting this theoretical approach [40] including studies in sports settings e.g., [41]. Consequently, EI could be expected to play a major role in boosting recovery processes and helping to protect, build, refill or restore personal resources when individuals have faced stressful situations.

Based on previous studies in extreme situations, the interplay between recovery and stress states has been shown to play a major role in the psychological adaptation processes. However, some gaps remain in the research. Specifically, the role of trait EI in the recovery process has not yet been investigated whereas promising theoretical support exists for the link between EI and recovery [39]. Consequently, the present study aims to provide experimental evidence regarding the psychological adaptation in MUM runners by investigating the recovery stress process and the relationship between recovery stress states and trait EI, especially before, during, and after one of the most extreme MUM races. This study could provide insights on how stress states could be tolerated and/or buffered in MUM in regard to recovery strategies [7]. Considering previous results on recovery stress states in ultra-marathons [14,15], it was hypothesized that (1) stress states would increase during and after the race compared to pre-race, whereas (2) recovery would decrease in the same time evolution. Furthermore, based on previous research on EI [2], we hypothesized that: (3) athletes with low trait EI would report lesser recovery and higher stress states during the MUM race than athletes with a high trait EI.

## 2. Methods

### 2.1. Participants

Thirteen athletes running an extreme MUM event (2 women and 11 men), aged from 29 to 52 years old (Mage = 40.08 yrs, SDage = 6.76), voluntarily participated in this study. Initially, 17 participants were recruited through an announcement for this study, which was part of a larger research project that focused on the physiological consequences of this race, which is considered to be the most challenging mountain ultra-marathon in the world. Of the 17 participants initially enrolled in the study, 13 completed the race (inclusion criterion) and thus constituted our sample for this study. The MUM was the Tor des Géants^®^ (TdG) and consisted of a semi-self-sufficiency race where the runners covered a total of 338 km with a cumulated altitude of 30,959 m of positive elevation under changing climate conditions (temperatures between −9 and 15 °C). For the rest, rescue, and refreshment points, runners were able to rely on the seven base camps spaced approximately 50 km apart. On average, participants accomplished the race in 132.67 h (*SD* = 13.16). All participants signed a consent form stipulating their right to withdraw from the experiment at any time without prejudice. This study was approved by the local ethics committee in accordance with the Declaration of Helsinki (amended 2013).

### 2.2. Measures

#### 2.2.1. Brief Emotional Intelligence Scale (BEIS-10)

The BEIS-10 [42] is based on both the EI model of Salovey and Mayer [6] and the work of Lane et al. [43]. The BEIS-10 was administered to athletes to measure their trait EI using a 6-point Likert scale (1 = never to 6 = always). The 10-item version is a short and efficient measure to quickly assess an individual’s perception of the extent to which they appraise, regulate, and use emotions. For this study, the internal consistency of the BEIS-10 was 0.84.

#### 2.2.2. Recovery Stress States (RestQ-36-R-Sport)

Based on the original Rest-Q for athletes [7], the French version of the RestQ-36-R-Sport questionnaire [18] was used to quickly assess the multidimensional nature (physical, emotional, behavioral, and social) of recovery and stress states. This questionnaire was developed to quickly measure the frequency of current stress along with the frequency of recovery using a 6-point Likert scale (1 = never to 6 = always). Higher scores in the stress responses reflected an intense and elevated perceived stress state. Higher scores in the recovery strategies reflected a high frequency of using numerous recovery strategies. Pre-TdG instructions given to participants for the completion of the RestQ-36-R-sport referred to «the 3 last days» whereas Per- and Post-TdG instructions referred to «the 3 last hours» in order to evaluate the psychological states during the MUM. The internal consistency for total stress and recovery scores across the several measurement times ranged from 0.53 to 0.98. Cronbach alpha tends to increase with an increase in the number of participants [44], leading researchers to suggest a cut-off value of 0.60 for a low sample size [45]. Other researchers prefer the use of the raw average inter-item correlation (AIIC) as a statistical marker of internal consistency. For this, a rule of thumb is offered by Clark and Watson [44] who recommend AIIC scores higher than 0.15. In the present study, all the AIIC scores were higher than 0.15.

### 2.3. Procedure

The thirteen participants rated their recovery stress states and trait EI on self-report questionnaires in the 3 h before the race (Pre-MUM). Secondly, while the majority of research on emotion, EI, or recovery stress states conducted in ultra-endurance sports has focused on the differences pre- and post-race e.g., [13,14,15], participants in the present investigation also rated their recovery stress states during the race. This measure was completed at the Donnas camp (located at the mid-race point: Per-MUM) after athletes had run 155 km with an average running time of 51 h (*SD* = 3.41). Thirdly, the athletes completed the RestQ-36-R-sport questionnaire for the last time during the three-hour period after finishing the race.

### 2.4. Statistical Analysis

Shapiro–Wilk and Levene’s tests were used to verify data normality and the homogeneity of variances at each time point. Trait EI data were analyzed: (a) using correlational analysis with recovery stress states scores on the total sample; and (b) by dividing its scores into either a high or low group based on the median value, a common practice within the literature [46,47,48]. While dichotomization is sometimes criticized in the literature [49], recent re-evaluations have shown that this practice is a robust, reliable, and appropriate statistical analysis when independent variables are uncorrelated [46,47,48]. A median split was therefore used to dichotomize participants based on their scores of trait EI. The results from our sample showed that the low trait EI scores and high trait EI scores were uncorrelated (*p* = 0.07) and provided evidence supporting the use of a median split in the present study. Literature has also shown that conducting a median split does not increase the likelihood of a Type I error [47]. Additionally, given that the scores for all factors at the different time measures were normally and homogeneously distributed, we conducted a set of multivariate analyses of variance (MANOVA) with repeated measures to test: (1) The effect of time on recovery stress states; (2) the effect of trait EI groups (high EI vs. low EI); and (3) the effect of the interaction of trait EI-groups * time. Follow-up univariate one-way ANOVAs were conducted in order to target significant differences detected using MANOVA. Pairwise comparisons (post hoc) were conducted using Tukey’s HSD.

## 3. Results

### 3.1. Descriptive Analyses

Descriptive statistics for recovery stress states and trait EI are shown in Table 1. Results of correlational analysis for the total sample showed that recovery was negatively correlated with stress state (*r* = −0.59, *p* < 0.05) whereas trait EI was not significantly correlated with stress state and recovery state.

Based on the median value (*Me* = 45), a significant difference between the high trait EI and low trait EI groups was observed in this study, *t*(11) = 4.53, *p* = 0.008, *d* = 2.23. The high EI-level group contained 6 athletes (*M*_age_ = 50.12; *SD* = 2.71) while the low EI group contained 7 athletes (*M*_age_ = 39.19; *SD* = 5.35). High trait EI and low trait EI mean scores were not significantly correlated (*p* > 0.05), encouraging the use of a median split. For the high trait EI group, correlations revealed that recovery was negatively correlated with stress state (*r* = −0.91, *p* < 0.05) and stress state was negatively correlated with EI (*r* = −0.90, *p* < 0.05). For the low trait EI group, no significant correlation was found.

### 3.2. Stress State

Figure 1 presents changes in stress scores during the TdG^®^. Firstly, the effect of the EI group on stress state was not significant, *F*(1, 11) = 0.74, *p* = 0.408 (*M low EI* = 2.77, *SD* = 0.13; *M high EI* = 2.62, *SD* = 0.14). Secondly, stress state scores changed over time, *F*(2, 22) = 5.19, *p* = 0.014, η_p_^2^ = 0.28. Tukey post hoc tests revealed significant increases, specifically between pre-MUM (*M* = 2.57, *SD* = 0.11) and post-MUM (*M* = 2.88, *SD* = 0.12, *p* = 0.017, *d* = 3.04) and between per-MUM (*M* = 2.62, *SD* = 0.11) and post-MUM (*p* = 0.043, *d* = 2.35). Thirdly, no significant interaction was observed, showing that stress scores were not influenced by the athletes’ EI levels throughout the race, Wilk’s λ = 0.97, *F*(2, 22) = 0.269, *p* = 0.767.

### 3.3. Recovery State

Figure 2 presents the changes in recovery states during the TdG^®^. Firstly, results showed no significant effect of the trait EI group on recovery scores, *F*(1, 11) = 3.77, *p* = 0.078 (*M low EI* = 3.31, *SD* = 0.16; *M high EI* = 3.78, *SD* = 0.17). Secondly, the effect of time on recovery scores was significant, *F*(2, 22) = 7.50, *p* = 0.003, η_p_^2^ = 0.45. Tukey HSD post hoc tests showed that the score for recovery decreased between pre-MUM (*M* = 3.81, *SD* = 0.12) and per-MUM (*M* = 3.46, *SD* = 0.09, *p* = 0.02, *d* = 3.11) and between pre-MUM and post-MUM (*M* = 3.43, *SD* = 0.09, *p* = 0.003, *d* = 3.58).

Thirdly, the interaction effect of trait EI group X time on recovery was significant, *F*(2, 22) = 12.21, *p* = 0.0003, η_p_^2^ = 0.53. The low trait EI group reported a lower score for recovery (*M* = 3.26, *SD* = 0.60) compared to the high trait EI group (*M* = 4.27, *SD* = 0.39, *p* = 0.004, *d* = 4.35) at Pre-MUM and this effect was non-significant at per-MUM and Post-MUM. In addition, only recovery scores in the high trait EI group decreased over time. Specifically, recovery scores decreased between pre-MUM (*M* = 4.27, *SD* = 0.19) and both per-MUM (*M* = 3.61, *SD* = 0.16, *p* = 0.0009, *d* = 3.11) and between pre-MUM and post-MUM (*M* = 3.46, *SD* = 0.19, *p* = 0.0002, *d* = 3.11) among the high trait EI group whereas no significant difference was observed among the low trait EI group. Finally, all participants during the race reported high recovery levels compared to stress states, *F*(2, 48) = 7.74, *p* = 0.001, η_p_^2^ = 0.24 (Table 2). A Tukey’s HSD post hoc test confirmed that all recovery scores were higher than the stress scores, either before, during, or after the race (Table 1).

## 4. Discussion

The purpose of this study was to examine the time courses of recovery stress states before, during, and after one of the most challenging MUM races. This study also contributes to identifying how individuals’ high versus low trait EI affects their recovery stress states. Results showed an imbalance between recovery stress states, highlighting an increase in the stress states and a decrease in the recovery states. This confirms the first two hypotheses and reaffirms that running a MUM race is a psychologically demanding situation. However, a particularly interesting finding concerns the differences in recovery states based on trait EI scores. As expected in regard to the third hypothesis, athletes with higher trait EI scores reported higher recovery states compared to athletes with lower trait EI scores. Our findings support the positive role of trait EI on an individual’s ability to cope with challenging situations [38].

Consistent with previous research on extreme situations [14,21], the results of the present study tend to reaffirm that the stress state is increased over time regarding an ultra-endurance race. Specifically, stress states significantly increased immediately after the race compared to the start of the race, while no significant variation of stress states was observed between pre- and per-MUM. Even if athletes tended to experience a constant stress state during the first part of the race, prolonged and repeated exposure to stressful environmental conditions increased the stress state after the race. It is well established that runners completing a MUM have to push their resources beyond ordinary limits [12] to cope with the severe demands placed upon them, such as physical repercussions (e.g., fatigue, sleep deprivation) [50], emotional disturbances [51], and social stress [14].

Athletes reported higher scores of recovery than stress at every time point. These results suggest that the recovery strategies were frequently used to buffer stress states. It seems that athletes who finished the race tended to efficiently manage their resources throughout the race. Based on their higher scores of recovery compared to stress, they had to prioritize recovery to ensure their performance, health, and well-being [52] an effective biopsychosocial adjustment during the race. However, it is also noteworthy that recovery decreased over the duration of the race, reflecting the difficulty of the runners in maintaining and using strategies to preserve physical and psychological resources throughout the race. The impairment of this balance may be explained by the fact that individuals have to draw on their own resources to achieve their goals over a long time [23]. In line with the findings for prolonged exposure to stressors, where recovery decreases and stress increases simultaneously [21], continuous effort in demanding situations could lead to exhaustion of psychological resources and in turn could prevent the use of recovery strategies [14].

Based on the median split approach, two groups were distinguished with significantly different trait EI scores. Although a significant negative correlation was observed between recovery and stress scores among the total sample, the correlation between recovery and stress scores only remained significant among participants belonging to the high trait EI group. As expected, recovery and stress states were significantly related in individuals reporting greater levels of trait EI. As suggested by Jeffrey [26], recovery stress states could be more balanced in an athlete with a high trait EI in order to find an optimal recovery within any challenge. Our results agree with this statement: Athletes who reported a high trait EI reported more recovery strategies (i.e., active, passive, and proactive), which could provide them with better control of their stress states before the race.

Surprisingly, scores for stress states were not statistically different between the high and low trait EI groups. Literature suggested that EI was associated with significantly lower stress scores in stressful situations (i.e., competition), highlighting the protective role of trait EI within stressful events [2]. However, our results do not confirm this literature. This could be explained by the potentially positive impact of the stress states, which may lead to psychological adaptation and coping within stressful environmental conditions. Stressful conditions actually lead to an increase in the stress responses in ultra-endurance athletes [13,14,15]. However, a certain level of stress state may be necessary for a successful psychological adaptation, as long as the recovery is sufficient to help mobilize personal resources [7,17]. Stress is therefore no longer considered to be a negative consequence because it supports adjustment. Thus, the goal is not to eliminate the stress state per se but rather to use it, while maintaining high scores of recovery, to buffer, manage, and regulate stress. In this way, trait EI could play a protector role in stress through cognitive appraisals in helping individuals evaluate situations as being challenging [53]. Reaching a balance between stress state and recovery state would be a particularly relevant strategy to promote adjustments in a MUM situation. In addition, the stress state experienced by athletes could be considered as eustress to help further increase and mobilize their personal resources in a constant adjustment to the extreme situation [23]. As a reminder, all participants in this study were part of the 55% of finishers, suggesting that an optimal recovery stress state was observed.

As expected, athletes who reported a high trait EI showed higher scores of recovery before the race. In other words, these runners tended to be more able to protect, build, refill or restore their personal resources compared to the low trait EI runners. This finding highlights the positive role of trait EI on the passive, active and proactive approaches to recovery, in addition to its positive influence on the use of several psychological skills, such as self-talk, imagery, or activation [43]. Our findings at the outset of the competition highlight that trait EI would help to optimize psychological processes by buffering stressor effects [28] and boosting personal resources (physical, emotional, behavioral, and social) [18]. However, an alternative explanation could be provided for the fact that only recovery scores in the high trait EI group decreased over time. High trait EI participants could have a better introspective sense of their internal state, whereas low trait EI participants may not have as fine-tuned a sense of their internal states and therefore did not report changes in their recovery states over time.

## 5. Limits

Due to limited access to the elite athlete population and finishers in these ecologically extreme situations, the small sample size of the present study represents a limitation for generalization and further analyses such as regression models. In this line, it would have been interesting to have more information on the characteristics of the participants to better understand our results by considering other potential biological, psychological, and sociological factors. For example, age, gender, and experience, but also training periodization, fitness, nutrition, and type of recovery practices may be involved in the development of recovery and stress states. Further research should consequently endeavor to recruit larger sample sizes and, more specifically, to go beyond the global EI score used in this study. This score was calculated from 5 distinct sub-dimensions: Appraisal of one’s own emotions, appraisal of others’ emotions, regulation of one’s own emotions, regulation of others’ emotions, and use of emotions. Previous research has revealed the relevance of investigating these sub-dimensions independently given that they could be differently associated with psychological responses, such as emotions [38,43]. Therefore, future research with a larger sample and different EI or emotional regulation questionnaires, e.g., CERQ [54]; PEC, [55]; TEIQue, [56] could lead to a more specific understanding of the respective influence of each dimension of trait EI on recovery strategies in stressful situations.

## 6. Practical Applications

This study gives insight into the role of trait EI in the recovery stress states during a MUM race. Runners should be aware that ultra-endurance races lead to substantial changes in recovery stress states and that trait EI could help them to improve their preparation for a MUM race. The ability to balance recovery stress states is essential in preventing pathogenic psychological outcomes but also for the development and maintenance of skilled performance, health, and well-being [1,19]. The positive association between trait EI and the recovery process could also help to improve pre-competitive resources and mental preparation. Coaches, athletes, and psychological counselors are concerned by this result because they could conduct specific interventions in order to improve the trait EI in athletes and in turn the balance between recovery strategies and stress states. As shown in previous studies, it is possible to improve trait EI [5,57]. EI interventions [58] should first focus on the understanding of the emotional information in order to lead individuals to be aware and accumulate sufficient knowledge to transform this into practice (i.e., recovery strategies) to increase trait EI.

## 7. Conclusions

Despite the limitations of this study, investigating the role of trait EI in MUM athletes should provide a better understanding of the balance of recovery and stress states. An added value of this study was to indicate that high trait EI was linked to higher scores of recovery before the race, suggesting that such athletes tend to be better prepared to cope with MUM. Athletes, coaches, and practitioners in sports psychology could develop trait EI in order to facilitate the use of recovery strategies and optimize personal resources in competition.

## Figures and Tables

**Figure 1 ijerph-19-09290-f001:**
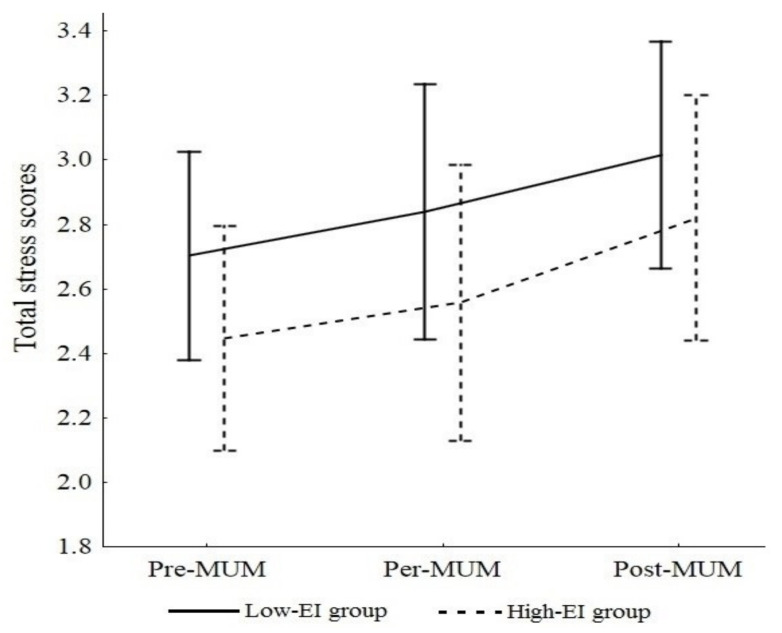
Total stress scores during MUM in high and low EI groups.

**Figure 2 ijerph-19-09290-f002:**
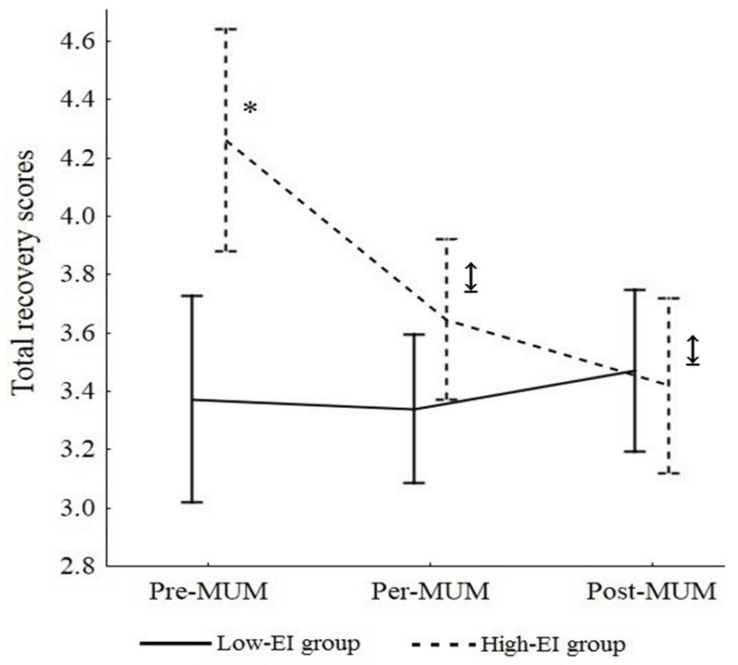
Total recovery score during MUM in high and low trait EI groups. Notes. * Mean of recovery for the high trait EI group significantly higher than the mean of recovery for the low trait EI group. ↨ Mean of recovery for the high trait EI group significantly higher at pre-MUM than per- and post-MUM and then the mean of recovery for the low trait EI group.

**Table 1 ijerph-19-09290-t001:** Descriptive statistics and inter-correlations for recovery stress states and EI scores in high trait EI (*n* = 6) and low trait EI (*n* = 7).

	Recovery	Stress	Emotional Intelligence
Total sample			
Recovery	-		
Stress	−0.59 *	-	
Emotional Intelligence	0.41	−0.25	-
*M*	*3.57*	*2.70*	*44.31*
*SD*	*0.47*	*0.30*	*6.79*
High trait EI group			
Recovery	-		
Stress	−0.91 *	-	
Emotional Intelligence	0.79	−0.90 *	-
*M*	*4.26*	*2.61*	*49.96*
*SD*	*0.17*	*0.14*	*3.03*
Low trait EI group			
Recovery	-		
Stress	−0.56	-	
Emotional Intelligence	0.18	0.14	-
*M*	*3.37*	*2.85*	*39.57*
*SD*	*0.16*	*0.13*	*5.86*

Note. * *p* < 0.05.

**Table 2 ijerph-19-09290-t002:** Results of the MANOVA analysis for recovery stress states in the low EI and high EI groups.

	Low Emotional Intelligence (*n* = 7)			High Emotional Intelligence (*n* = 6)			Tukey’s HSD Interpretation
	Pre-MUM (1)	Per-MUM (2)	Post-MUM (3)	Pre-MUM (4)	Per-MUM (5)	Post-MUM (6)	
	M(SD)	M(SD)	M(SD)	M(SD)	M(SD)	M(SD)	
Recovery (R)	3.30 (0.23) *	3.32 (0.11) *	3.48 (0.14) *	4.02 (0.23) *^µ^	3.57 (0.11) *	3.40 (0.14) *	
EI-level effect	*F*(1, 11) = 5.23, *p* = 0.04, η_p_^2^ = 0.322						R in high EI > R in low EI
Time effect	*F*(2, 22) = 9.10, *p* = 0.001, η_p_^2^ = 0.452						R at Pre-MUM > R at Per- and Post-MUM
EI level * Time	*F*(2, 22) = 12.53, *p* = 0.0002, η_p_^2^ = 0.532						1 < 4; 4 > 5–6
Stress (S)	2.70 (0.15)	2.84 (0.18)	3.01 (0.16)	2.45 (0.16)	2.56 (0.19)	2.82 (0.17)	
EI-level effect	*F*(1, 11) = 1.53, *p* = 0.24, η_p_^2^ = 0.122						NS
Time effect	*F*(2, 22) = 4.36, *p* = 0.03, η_p_^2^ = 0.283						S at Pre-MUM < S at Post-MUM
EI level * Time	*F*(2, 22) = 0.07, *p* = 0.93						NS

Note. * Mean of recovery significantly higher than mean of stress. NS = non-significant; ^µ^ Mean of Pre-MUM recovery in high EI group significantly higher than Mean of pre-MUM recovery in low EI group.

## Data Availability

The data are available and can be sent by the corresponding author.

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
