# Peer review of "Emotional Intelligence in Ultra-Marathon Runners: Implications for Recovery Strategy and Stress Responses during an Ultra-Endurance Race"

_ijerph, 2022, doi:10.3390/ijerph19159290_

Round 1

Reviewer 1 Report

This an exceptionally well written manuscript reporting on the results of a study investigating the influence of emotional intelligence on recovery and stress states among ultra-runners. The theoretical underpinnings of the research are evident throughout the introduction, and the rationale for the hypotheses was effectively established. The methodological decisions seem justified, as does the statistical analyses that were conducted. The results are clearly reported, and the interpretation of the results in light of previous research was thorough. I think the practical implications of the research are clear, and I concur with the authors' conclusions regarding the importance of exploring interventions to improve emotional intelligence; not just among ultra-runners, but in other contexts as well. My only concern is the use of the term experiment or experimental (e.g., lines 57, 168, and 189) being this appears to have been an observational study. Even though it was in situ, there was no attempts to manipulate any study variables or conditions by the investigators. I will defer to the editor whether this is a problem, as I humbly admit I may not understand the criteria for experimental designs in other disciplines (my background is exercise physiology and exercise psychology). I also think there is room to mention, at least in the limitations, there is much we don't know about the participants that could partially explain differences  in recovery and stress states (e.g., nutritional status, hydration status, injury status, etc.). 

Reviewer 2 Report

Thanks for the opportunity for reading your work. I'm approaching this review from the perspective of a researcher with experience in negotiating the understanding and application of EI, with a passion for evidence quality. I do hope the following comments are of benefit.

 - Your positioning of emotional intelligence is very broad and could do much more to highlight the approach adopted to avoid broad overgeneralisations. For example in previous work (Hughes & Evans, 2019) I have differentiated between ability, trait and emotion regulation approaches to EI. It would be great to get clarity on which you are referring to and to see alignment in your positioning, the approach adopted by the literature you adopted, your model and measurement approach, and your conclusions. This is seen throughout your work e.g. in your discussion when you discuss interventions to increase EI - this is only plausible for certain approaches to EI.

- It would be really useful to have more detail on how participants were sampled/recruited/selected and how that process was managed.

- The choice of EI measure is somewhat questionable given that it is not clearly aligned to a specific approach to EI. Furthermore, the questions are not especially well-focused and could be considered to capture a wide-range of different individual differences e.g. "I arrange events others enjoy". I appreciate the need for brief measures but I would be very cautious about any conclusions based upon this measure.

- Dichotomisation of variables like EI is highly problematic and the citation provided to support this practice is from a paper in the field of consumer psychology and is likely to be inappropriate to apply in this circumstance. This practice means any nuance in the data has been lost and increases the likelihood of making erroneous or misleading conclusions. On this basis, I would be concerned that the analytical procedure adopted is suboptimal.

- The data visualisation strategy adopted is somewhat suboptimal and it might be worth considering whether you can better represent error/confidence in the visualisations to help convey more of the nuance behind the conclusions provided.

- Research transparency is generally low and without this it becomes very difficult to effectively evaluate the work fully. I encourage you to share the materials and data on stable platforms (e.g. the OSF.io) and including a link to these in the manuscript, before any future submissions.

- The discussion could provide a more effectively nuanced consideration of the study. For example, the sample size is small (understandable given the population being studied) but the implications of this for our confidence in your findings could be discussed with much greater nuance.

In sum, this is really interesting work and I am grateful for the opportunity to read it, however I am concerned by the discussion, operationalisation, measurement, analysis, and discussion of EI throughout this work, and there are a number of key factors (e.g. sample size, analysis approach, etc.) which would prevent me from completely trusting and endorsing this work. I appreciate much of this is likely to be disappointing, however I hope this feedback can be useful to you and I wish you all the very best,

*Please note that I do not comment on the novelty of research as this is not a rigorous or valuable dimension by which to consider research. As such, please do not see my 'no answer' response as either endorsing or critiquing the research submitted. 

Round 2

Reviewer 2 Report

Thanks for the opportunity to revisit your manuscript. Unfortunately, many of my core concerns surrounding the work remain. For example, there may be some acknowledgement that EI is multifaceted but there is still not coherence between the model adopted, the measure, and the conclusions surrounding it. My concerns surrounding the splitting of a continuous variable into a dichotomous variable was not resolved by the inclusion of two references in sport contexts - this practice is still problematic and likely to lead to erroneous conclusions. Finally, the data and materials for the study have not been shared. As such, hopefully you can see why I do not believe my concerns have been fully actioned. I appreciate these are quite fundamental components of the project and so they are hard to change, but that is also why it is so important that they are changed to ensure the scientific record is clear and impactful. I really hope these comments help and I wish you all the very best,

Stay safe and take care.
